# Plant different, eat different? Insights from participatory agricultural research

**Carlo Azzarri**[1], **Beliyou Haile**[1]*, **Marco Letta**[2]

**1** International Food Policy Research Institute (IFPRI), Washington, DC, United States of America, **2** DiSSE —Department of Social Sciences and Economics, Sapienza University of Rome, Rome, Italy

* b.haile@cgiar.org













**Data Availability Statement:** All datasets are available from an online data repository platform called Dataverse: https://dataverse.harvard.edu/dataset.xhtml?persistentId=doi:10.7910/DVN/28557. However, datafiles on Dataverse are

## Abstract

We examine the association between on-farm production diversity on household dietary diversity in Malawi using microdata collected as part of an environmentally sustainable agricultural intensification program. The program primarily focuses on the integration of legumes into the cropping system through maize-legume intercropping and legume-legume intercropping. Relative to staple cereals such as maize, legumes are rich in micronutrients, contain better-quality protein, and lead to nitrogen fixation. Given the systematic difference we document between program beneficiaries and randomly sampled non-beneficiary (control) households, we employ causal instrumental variables mediation analysis to account for non-random selection and possible simultaneity between production and consumption decisions. We find a significant positive treatment effect on dietary diversity, led by an increase in production diversity. Analysis of potential pathways show that effects on dietary diversity stem mostly from consumption of diverse food items purchased from the market made possible through higher agricultural income. These findings highlight that, while increasing production for markets can enhance dietary diversity through higher income that would make affordable an expanded set of food items, the production of more nutritious crops such as pulses may not necessarily translate into greater own consumption. This may be due to the persistence of dietary habits, tastes, or other local factors that favor consumption of staples such as maize and encourage sales of more profitable and nutritious food items such as pulses. Pulses are a more affordable and environmentally sustainable source of protein than animal source food, and efforts should be made to enhance their nutritional awareness and contribution to sustainable food systems and healthier diets.

## Introduction

Agricultural interventions in most of Africa historically had narrow focus on increasing yield per unit area of a few staple crops through labor-saving (e.g., mechanization) and land-saving (e.g., fertilizer and pesticide) technologies without adequate consideration for the whole ecosystem and implications for nutrition and health [1,2]. In light of persistent food insecurity and climatic risks the region is facing, interest has grown on how best to leverage agriculture to tackle undernutrition while simultaneously improving the natural resource base and

currently locked since they are under audit for conformity with donor's open access requirements. They will be made accessible to the public once the audit is completed. We are also happy to upload a subset of the datasets used in this study on other platforms if the paper is accepted but the audit is still ongoing.

**Funding:** This research was conducted as part of the Africa Research In Sustainable Intensification for the Next Generation (Africa RISING) Program, funded by United States Agency for International Development as part of the US Government's Feed the Future Initiative [grant number #AID-BFS-G-11-00002]. The donor had no role in study design, data collection, analysis and interpretation of results, decision to publish, or preparation of the manuscript.

**Competing interests:** The authors have declared that no competing interests exist.

resilience. A recent review paper from Africa and other developing regions shows that most of the evidence supports a positive association between adoption of agroecological practices (e.g., crop diversification, cereal-legume intercropping, agroforestry, crop-livestock integration, and integrated soil and water management practices) and food security [3].

Agriculture contributes to the food security of poor rural households both directly, by boosting food availability for subsistence-oriented farmers, and indirectly, by enhancing income for commercially oriented farmers. Diversification of agriculture production into nutrient-rich crops and animal-sourced foods (ASF) is often considered as one of the options for improving diets and nutrition among smallholders considering their reliance on own-produced foods [4]. Especially when access to markets for buying food and selling agricultural production is limited, diverse agricultural production can play a vital role in ensuring diversified food consumption [5]. For example, evidence from Ethiopia shows that households that live far away from market centers not only consumed less diverse foods but also had smaller food consumption expenditure relative to households who live close to markets [6].

Most of the empirical studies on the linkages between on-farm diversity and diets as well as the role of mediating factors such as access to food markets rely on cross-sectional data to generate evidence about associations versus causality [7–16]. The evidence generated from these studies is mixed not only across studies but also within a study based on geography, diversity indicators, and commodities [4,16–18].

Dependence on cross-sectional data poses several challenges while establishing linkages between production diversity and dietary diversity due to a host of confounding factors that jointly and simultaneously affect production and consumption decisions. This study uses cross-sectional data from Malawi to examine linkages between on-farm production diversity and household dietary diversity. We contribute to the literature by analyzing household level dietary patterns among households who participated in an agricultural intensification program that aims to integrate pulses into Malawi's maize-dominated farming systems.

Taking advantage of quasi-experimental agricultural and dietary data collected from program participant and non-participant (control) households, we use the recently developed mediation analysis approach applied to instrumental variables (IV) frameworks [19,20] in which multiple endogeneity and transmission mechanisms are simultaneously at play and are controlled for using a single instrument. The approach has been successfully employed in several recent publications [21–23].

We find that participating into the program leads to a significant increase in production diversity that, in turn, translates into more diverse diets. The main underlying mechanism, however, appears to be working through higher purchase of more diverse foods, rather than through an increase in own consumption of pulses, the main crops targeted by the program. Finally, these effects are primarily due to an increase in the production of a secondary targeted crop, maize, and not to higher pulse production. These findings highlight that, while increasing production for markets can lead to more diverse diets through increased income that would make affordable an expanded set of food items, increasing production of more nutritious crops such as pulses may not necessarily translate into greater own consumption due to the persistence of dietary habits, tastes, or other local factors that favor consumption of staples such as maize and encourage sales of more profitable and nutritious food items such as pulses.

## Study setting

Malawi's crop production is highly dominated by continuous maize production. For example, a recent study based on a panel of field-level data from Central and Southern Malawi showed maize was continuously grown for four years on more than 55% of the plots in Central region

and more than 82% of the plots in Southern region [24]. This practice has implications not only for depletion of essential soil nutrients but also for the diversity of household diets. Through Malawi's Farm Input Subsidy Program (FISP), the government has been providing subsidies to smallholders for inorganic fertilizers and improved seeds for maize since 2004/05. The provision of improved seeds as part of FISP was expanded to legumes in 2008/09 in recognition of their contribution for improving soil quality and nutrition [25]. Nonetheless, maize is still Malawi's main staple food accounting for about half of the total plant-based caloric intake with pulses and groundnut accounting for just 7% [26].

More than 90% of Malawi's population lives under $1.9 a day poverty line (in purchasing power parity), 19% of the population is undernourished, and 40% of children below five years old are stunted [27]. Amid evidence of FISP failing to meaningfully reduce food insecurity and enhance dietary diversity, interest has grown on how best to leverage various agricultural strategies to effectively tackle undernutrition including through adoption of agroecological principles and diversification [5,28–30]. While crop diversification is among the goals of Malawi's agricultural sector development strategy, ensuring maize self-sufficiency remains the main focus of public agricultural investments [31]. Adoption of agroecological practices in Malawi has been linked with improved food security [30] and stronger association between on-farm and dietary diversity [28]. A positive association has also been documented between crop diversification and dietary diversity with varying magnitude [5,9,29].

This study is conducted as part of an agriculture research for development program called Africa RISING being implemented in Dedza and Ntcheu districts of Malawi since 2012. The program aims to promote sustainable intensification primarily through integration of legumes (groundnut, pigeon pea, cowpea, soybean, and beans), organic and inorganic fertilizers and livestock innovations. Through a participatory 'mother-and-baby' trial design [32], the program has been validating integrated packages of agricultural technologies for subsequent scaling. Interactive and replicable demonstration ("mother trial") plots were established around farmer action groups, whose members subsequently set up adaptive ("baby trial") plots to test a subset of technologies from the mother trial (hereafter program beneficiaries). Fig 1 shows the mix of agricultural innovations tested by program beneficiaries at the time of the baseline where pulses -with or without maize and with or without inorganic fertilizers- were quite common.

One of the criteria for setting up baby trials was that farmers shall select no more than four (integrated) technologies and be able to devote at least 10 square meters of accessible land for each treatment chosen. As shown later, beneficiaries appear to be systematically different from randomly selected control households including cultivating bigger land area. Significant attention was given to the practice of intercropping, namely maize-legume intercropping or intercropping of two legumes with different growing periods-known as doubled-up legume technology [33]. Compared to cereals, pulses are rich in crucial micronutrients, contain better and higher quality protein, can be a more affordable source of protein compared to ASF. Legume intercropping helps improve soil fertility, yield, and nutrition while reducing fertilizer requirements due to nitrogen fixation [34–36]. Various efforts are underway to promote pulse production and their multifunctional roles within the smallholder farming systems [37–39].

## Materials and methods

### Data and key variables

Microdata analyzed here were collected as part of a baseline survey approved by the Institutional Review Board of the International Food Policy Research Institute (IRB # 00003487). Written informed consent was obtained from all survey participants. Data were collected

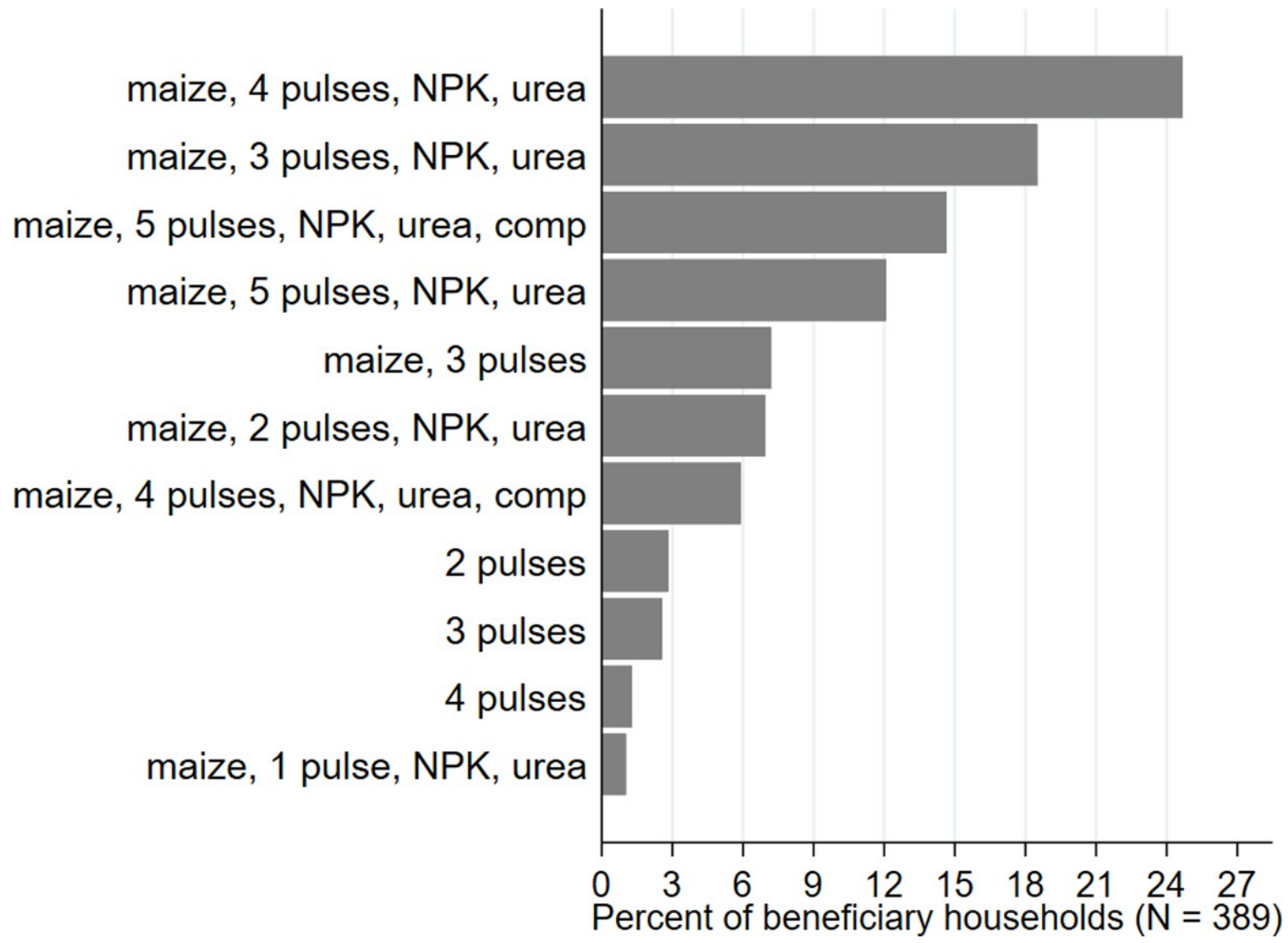


**Fig 1. Mix of innovations tested by program beneficiaries.** Note: The five pulses include groundnut, pigeon pea, cowpea, soybean, and common bean; *NPK*: Nitrogen, phosphorous, and potassium; *comp*: Compost.


between August and October 2013 after program beneficiaries obtained their first harvest since joining the program [40]. As reported in a previous study [41], detailed agricultural and socio-economic data were collected from two groups of households: 1) all program beneficiary 'baby' farmers -and their households- that were engaged in testing program technologies as of June 2013 (N = 397) and 2) randomly sampled (control) households drawn from non-program target villages with comparable biophysical and agro-ecological conditions as program target villages (N = 538). Agricultural production data refer to 2012/13 long rainy season while food consumption data are based on 7-day recall period.

Production diversity was largely captured at the household level based on crop and/or livestock species and alternative indicators (e.g., count of unique food items, count of food groups with similar nutritional content, indicators of crop species richness and/or evenness) while dietary diversity was measured either the household or individual level based on metrices that include dietary diversity scores, food variety scores, and weighted food consumption scores [18]. In this study, we adopt a good group approach to measure diversity.

First, we classify all food items into 12 groups [42]: cereals, roots and tubers, pulses and nuts, vegetables, fruits, meat, eggs, fish, milk and dairy products, oils and fats, sugar and sweets, and miscellaneous items including spices, condiments, and beverages. Next, we compute total food consumption expenditure as the sum of expenditures on purchased food and imputed values of food consumed from own production and gifts. Imputation is based on food item-specific unit price values [43] computed by dividing total expenditure on purchased food by total quantity of purchased foods. We performed an outlier check and corrected for outliers in food value by replacing monetary values higher than +3 Standard Deviations (SD) or lower than -3 SD from the median by the distribution median. Finally, we construct Simpson's diversity index [44], defined as $1 - \sum_1^{12} (e_j/e)^2$, where $j$ is food group, $e_j$ is annualized household food expenditure on $j$ in local currency -Malawian kwacha (MWK)-, and $e = \sum_1^{12} ej$.

Simpson's diversity index measures both richness -i.e., the number of food groups- and relative abundances -i.e., the extent to which food expenditure is uniformly spread across food

groups- with values ranging between zero and one. It represents the probability that two randomly selected food items belong to two different food groups [45], with its value increasing with the number of food groups consumed, the evenness of household budget share distributions, or both. To assess the diversity of purchased foods, we construct an additional Simpson's index based only on food items that were purchased.

Similarly, we construct Simpson's on-farm production diversity index based on self-reported data on production of crops, livestock, and animal by-products. Production diversity is based on the following nine food groups: cereals, roots and tubers, pulses and nuts, vegetables, meat, eggs, fish, milk and dairy products, and cash crops such as sugarcane, cotton, and tobacco. Similar to the approach used for food consumption, we compute unit values of agricultural commodities by dividing total sales revenue by total quantity sold and subsequently use them to monetize the total value of agricultural production. Outliers in the value of agricultural production are corrected by replacing values higher than +3 SD or lower than -3 SD from the median by the median. As a sensitivity analysis, we estimate models of production and dietary diversity using Shannon's diversity index. While Simpson's index is a dominance index that assigns more weight to dominant food groups, Shannon index emphasizes the richness component of diversity and is given by $-\sum_{j=1}^{J} e_j ln e_j$ where ln is natural logarithm and $e_j$ is as defined before.

In addition to diversity indicators, we construct several socioeconomic variables that may be correlated with program participation and agricultural production, and mediate the interactions between production and dietary diversity. These include household demographic characteristics, use of agricultural inputs and practices, and four standardized (with mean zero and standard deviation one) indices based on the number of durable agricultural assets (excluding land), the number of durable non-agricultural assets, the quality of the household's dwelling condition, and market access based on self-reported travel time to various services including the nearest markets (daily and weekly), nearest roads (motorized, all season, and asphalt), and schools. The indices are constructed using factor analysis based on principal-component factor method [46].

## Identification

Our primary identification strategy relies on IV-mediation analysis [19,20]. Mediation analysis is common in the social sciences outside economics, and is gaining prominence in applied economic research in recent years [47–49]. The goal of mediation analysis is to unpack the transmission mechanisms in which a treatment T and a mediator M jointly cause an outcome of interest Y, by disentangling the total effect of T (TE) on Y into two components: the indirect or mediated effect, that is the effect of T on Y that operates exclusively through its effect on M; and the direct effect, that is the residual effect of T on Y that is not mediated by M, that is holding the distribution of M constant.

The traditional mediation analysis assumes T to be randomly assigned, an assumption that does not hold in our case. Non-random treatment assignment makes T endogenous with respect to both M and Y. Additionally, in the model M is potentially endogenous to Y. While the standard IV framework can be used to address endogeneity bias in the form of non-random selection and reverse causality, it is not suitable to identify the causal effect of the mediator on the outcome of interest -the focus of this study. Furthermore, finding an instrument that is relevant and meets the exclusion restriction is empirically challenging, especially when two or more valid, exogenous, and strong instruments are necessary to isolate the causal effects as would be needed here.

Our empirical approach blends the potential of mediation analysis in disentangling the causal chain between different outcomes with the ability of the IV framework to tackle endogeneity, called IV mediation analysis [19,20]. The approach allows one to use the same instrumental variable to identify the complex chain among the outcomes within a standard IV strategy that controls for treatment endogeneity with respect to the intermediate and final outcomes. In particular, mediation effects can be identified when the IV model is partially confounded, that is when the unobserved confounding variables expected to affect the treatment and the intermediate outcome are independent of the confounders that affect the intermediate and final outcomes [50]. This is the case when T is endogenous in a regression of M on T due to confounders that jointly affect M and T, and T is endogenous in a regression of Y on T due to the same confounders that affect Y primarily through M.

The two stage least squares (2SLS) estimation procedure to identify the causal effect of T on M ($\beta_M^T$) can be formalized by the two-equation system shown in Eqs 1 and 2

$$\text{First Stage}: \ \mathrm{T} = \beta_{\mathrm{T}}^{\mathrm{Z}}\mathrm{Z} + \boldsymbol{\varepsilon}_{\mathrm{T}} \tag{1}$$

$$\text{Second Stage}: \ \mathrm{M} = \beta_{\mathrm{M}}^{\mathrm{T}}\hat{\mathrm{T}} + \boldsymbol{\varepsilon}_{\mathrm{M}} \tag{2}$$

where Z is the instrument and $\hat{T}$ is the predicted value of T from Eq 1. Under the identifying assumption for IV mediation analysis[20], causal effects of T on Y can be estimated via 2SLS estimation of Eqs 3 and 4 and a single instrument Z.

$$\text{First Stage}: \ \mathrm{M} = \gamma_{\mathrm{M}}^{\mathrm{Z}}\mathrm{Z} + \gamma_{\mathrm{M}}^{\mathrm{T}}\mathrm{T} + \boldsymbol{\varepsilon}_{\mathrm{M}} \tag{3}$$

$$\text{Second Stage}: \ \mathrm{Y} = \beta_{\mathrm{Y}}^{\mathrm{M}}\hat{\mathrm{M}} + \beta_{\mathrm{Y}}^{\mathrm{T}}\mathrm{T} + \boldsymbol{\varepsilon}_{\mathrm{Y}} \tag{4}$$

In Eq 4, the indirect or mediated effect of T on the outcome is provided by $\beta_Y^M$, the direct or residual effect of T on the outcome is given by $\beta_Y^T$, and the total effect is given by $\beta_Y^M + \beta_Y^T$. In our case, this approach allows us to investigate the production-consumption transmission chain while accounting for non-random selection into the program, endogeneity, and potential simultaneity between production and consumption decisions. In this complex transmission chain, we assess the effect of participation in the Africa RISING program (T) on the Simpson's household dietary diversity index (Y) as mediated by the effect of T on another mediator outcome (M), the Simpson's on-farm production diversity index. We posit the partial identifying assumption to hold in our setting, as it is plausible to assume that program participation is endogenous in a regression of dietary diversity (DD) on T only due to confounders that jointly affect T and production diversity (PD).

As the explicit goal of the program was to diversify agricultural production, there are only two main channels through which the treatment (participation) could affect the dietary diversity of beneficiary households: either by increasing their production diversity through the cultivation of new cash crops, thus raising their agricultural income and, in turn, their consumption of more diverse foods; or by diversifying the variety of subsistence crops and, in turn, directly boost their dietary diversity. Both these channels are mediated by agricultural production diversity. For these reasons, which are strictly related to the nature and design of the project, we can safely assume that T is endogenous with respect to DD due to confounders that affect DD primarily through PD.

Adapting the formalization of the IV mediation analysis framework to our study, we estimate Eqs 5 and 6 using 2SLS to examine the associations between program participation,

production diversity, and dietary diversity.

$$\text{First Stage}:\ \mathrm{PD} = \gamma_{\mathrm{PD}}^{\mathrm{Z}} * \mathrm{Z} + \gamma_{\mathrm{PD}}^{\mathrm{T}} * \mathrm{T} + \Phi'\mathbf{X} + \boldsymbol{\varepsilon}_{\mathrm{PD}} \tag{5}$$

$$\text{Second Stage}:\ \mathrm{DD} = \beta_{\mathrm{DD}}^{\mathrm{PD}} * \hat{\mathrm{PD}} + \beta_{\mathrm{DD}}^{\mathrm{T}} * \mathrm{T} + \Phi\prime\mathbf{X} + \boldsymbol{\varepsilon}_{\mathrm{DD}} \tag{6}$$

where PD and DD represent Simpson's production and dietary diversity indices, respectively; T is indicator for participation in Africa RISING program; Z is the instrument; **X** contains a set of control variables including household size, age and gender of the household head, average years of education in the household, dependency ratio, indices for non-agricultural assets and distance from basic services, temperature, slope, and indicators for self-reported shocks (drought and crop diseases); and $\varepsilon_{PD}$ and $\varepsilon_{DD}$ are model error terms. The estimate for the PD-mediated indirect effect of T on DD is given by $\beta_{\mathrm{DD}}^{\mathrm{PD}}$, the direct or residual effect of T on DD is given by $\beta_{\mathrm{DD}}^{\mathrm{T}}$, and the total effect T on DD is $\beta_{\mathrm{DD}}^{\mathrm{PD}} + \beta_{\mathrm{DD}}^{\mathrm{T}}$.

Our choice of instrument was guided by the fact that beneficiaries were more likely to operate plots closer to the homestead relative to the control group. A summary of plot location based on self-reported travel time data shows that 48% and 52% of beneficiaries' plots (N = 1,079) and 38% and 63% of control group plots (N = 1,008) were located, respectively, within 15 minutes of travel (nearby) and more than 15 minutes of travel (faraway) from the homestead. That is, the average plot owned by beneficiaries is more likely to be nearby while the opposite is true for the average plot owned by the control group. Average plot size was also statistically different by plot location–0.76 (0.9) hectares (ha) for nearby (faraway) plots for the whole sample and 0.73 (.92) ha for beneficiaries. Smaller nearby plots may be due to shortage of agricultural land in and near residential areas.

While we do no not find statistically significant differences in several agronomic indicators including yield by parcel location for the whole sample, we observe statistically significant differences in the number of crops grown per parcel and the use of cereal-legume intercropping by parcel location and treatment status. Specifically, and relative to nearby plots, beneficiaries grow higher number of crops per parcel (2.5 versus 2.3) and are more likely to practice cereal-legume intercropping (55% versus 45%) on faraway plots while the opposite is true for control households–where they grow 2.2 crops on faraway plots (versus 2 crops on nearby plots) where they also practice cereal-legume intercropping on 55% of the plots (versus 47% for nearby plots). The fact that beneficiaries needed devote at least 10 square meters to each program promoted technologies to participate in the program [41] and that faraway plots were, on average, bigger might explain the higher incidence of use of intercropping practice and number of crops on faraway plots.

We use total area of nearby plots to instrument for program participation and conduct two tests to provide indirect empirical evidence in favor of the exclusion restriction of our instrument. The first placebo test involves running a reduced form regression of the intermediate outcome–production diversity– controlling for the instrument, other covariates discussed above, and EPA fixed effects separately on overall, treated, and control group samples. We expect statistically significant coefficients of the instrument for the treated group but not for the control group. In the second placebo test, we test whether the exclusion restriction holds by estimating a reduced form model of the final outcome – dietary diversity– separately for overall, treated, and control groups. Once again, the coefficient of the instrument for the sub-sample of control households should not be statistically significant since the only way through which landholdings close to the homestead would affect dietary diversity is through production diversity to which the control group has not been exposed.

Since an IV estimator yields inconsistencies and finite-sample biases when the instrument (s) are weakly correlated with the endogenous variable(s), we conduct diagnostic tests of instrument relevance based on the significance of the excluded instrument in the first-stage reduced form regression [51]. With one instrument, the general rule of thumb is to reject the null of weak instrument if the F statistic is at least 10. As sensitivity analysis and to better understand impact pathway from program participation to household diet, we conducted three additional tests. First, we employ three different specifications of the system of Eqs (3) and (4). The first uses Simpson's dietary diversity index constructed based on purchased food consumed inside the household, instead of Simpson's diversity index based on all food consumed used in the main specification. The second and third specifications replace the production diversity indicator with the average value of maize, and legumes and nuts harvested per hectare, respectively. These alternative specifications allow us to assess the extent to which the effect of program participation on dietary diversity is driven by increase in the diversity of purchased food (as opposed to own consumption) and the contribution of program target crops.

Second, we estimate Eqs (3) and (4) using two household-level production and dietary diversity indices based on Shannon's diversity index constructed for agricultural production diversity and household dietary diversity. While the association between production and dietary diversity may depend on indicators used to measure diversity [17,18], several food-group based indicators of dietary diversity are found to be positively correlated with each other and with food and nutrition security [52].

Third, we use inverse probability weighting with regression adjustment (IPWRA) [53] to estimate average treatment effect on the treated (ATT) and average treatment effect (ATE). IPWRA addresses the endogeneity associated with self-selection into treatment by modelling both treatment selection and outcome variables rendering it a "doubly robust" estimator, meaning that either the treatment or outcome (but not both) must be correctly specified to consistently estimate treatment effects [54]. IPWRA is consistent if either the selection or outcome models are correctly specified and is more efficient, especially relative to weighting adjustment, if the outcome model is correctly specified [53,55]. The selection model controls for area of nearby plots (the instrument) and household characteristics discussed above while the linear model for dietary diversity controls for household characteristics, climatic variables, self-reported shock experience, as well as production diversity. Controlling for production diversity minimizes omitted variables bias in the and mimics the IV mediation analysis where the mediator is production diversity.

While the conditional mean independence assumption for matching is inherently untestable, we assess matching quality based on Rubin's bias (B) and ratio of variance (R) [56] and propensity score distributions before and after matching using box plots and density function. Rubin's B refers to the absolute standardized difference in the means of the propensity score between beneficiary and control groups while Rubin's R is measured as the ratio of variances of the propensity scores between the beneficiary and control groups. Rubin's R should be below 2 to avoid over-correction of bias and above 0.5 to prevent under-correction, while Rubin's B should be below 25. Robust (for IV mediation and reduced form models) and bootstrapped (for IPWRA) standard errors are reported.

## Results and discussion

Before presenting regression results, Table 1 compares selected socioeconomic and biophysical variables by treatment status mimicking balance t-tests of baseline variables. To recall, survey data were collected right after beneficiaries obtained their first harvest as a program beneficiary. But we maintain that variables reported in Panel A of Table 1 are unlikely to have been

**Table 1. Descriptive summary of socioeconomic variables by beneficiary status.**

| | (1) | (2) | (3) |
|---|---|---|---|
| | **Beneficiary** | **Control** | **Stat. sign.** |
| *Panel A. Sample characteristics* | | | |
| Household size | 4.97 | 4.59 | *** |
| Household head age (years) | 45.8 | 45.3 | |
| Female households (%) | 27.0 | 33.8 | ** |
| Average adult years of education (years) | 5.20 | 4.72 | *** |
| Number of adults (age> = 15) | 2.66 | 2.45 | *** |
| Number of children (age<15) | 2.30 | 2.13 | * |
| Land size (ha) | 1.20 | 0.86 | *** |
| Total land area within 15 minutes of travel | 1.18 | 0.51 | *** |
| Poor households based on durable agr. assets excluding land (%) | 29.5 | 40.7 | *** |
| Poor households based on durable non-agr. assets (%) | 26.4 | 40.7 | *** |
| Poor households based on dwelling condition (%) | 35.8 | 62.8 | *** |
| Livestock (Tropical Livestock Units) | 0.45 | 0.21 | *** |
| Distance to basic services index | 0.036 | -0.044 | |
| Remote households (%) | 34.5 | 32.3 | |
| *Panel B. GIS variables* | | | |
| Elevation (meters) | 864.6 | 945.6 | *** |
| Slope (degrees) | 1.25 | 0.98 | *** |
| Total annual rainfall (millimeters) | 931.5 | 919.2 | *** |
| Average monthly temperature (degree Celsius) | 21.4 | 21.0 | *** |
| *Observations* | *397* | *538* | *935* |

Note. Households are defined as poor if they fall in the lowest tercile of wealth index constructed based on durable agricultural (agri.) assets, durable non-agri. assets, or quality of dwelling condition. Households are defined as remote if they fall in the highest tercile of the index constructed based on travel time to various services. Columns 1 and 2 report means and column 3 reports statistical significance (stat. sign.) from mean comparison tests.

* $p<0.1$

** $p<0.05$

*** $p<0.01$.


affected by program participation given the short time lapse. Beneficiaries are more likely to have larger family size, be male headed, better educated, and less likely to be poor where poverty is measured based on durable agricultural and non-agricultural assets, and quality of dwelling conditions (Table 1, Panel A). Total land size operated as well as land size within 15 minutes of travel from the homestead -our selected instrument- are also greater for beneficiaries. We also observe some differences in biophysical conditions that may affect agricultural potential (Table 1, Panel B).

Table 2 presents descriptive summary of variables that are likely to have been affected by the program. Beneficiaries are more likely to use manure and hired labor; operate larger number of intercropped plots, and are more likely to have received agricultural extension information in the preceding year (Table 1, Panel A). Program beneficiaries also have better agricultural performance in terms of maize yield, total value of crop harvested, net agricultural income, on-farm diversity, and marketed surplus (Table 2, Panel B). Inter-group differences are also observed in term of the share of households producing different food groups as well as the monetary value of food groups produced (see supplemental S1 Table). These gains in the value and diversity of agricultural production appear to have been translated into higher food

**Table 2. Descriptive summary of agricultural and dietary outcomes by beneficiary status.**

| | (1) | (2) | (3) |
|---|---|---|---|
| | **Beneficiary** | **Control** | **Stat. sign.** |
| *Panel A. Agricultural inputs and practices* | | | |
| Agricultural labor used (person-days/ha) | 321.3 | 317.9 | |
| Household uses hired labor (%) | 49.9 | 39.0 | *** |
| Household uses communal labor (%) | 35.3 | 31.6 | |
| Inorganic fertilizers applied (kg/ha) | 114.1 | 103.3 | |
| Number of intercropped plots | 1.88 | 1.16 | *** |
| Received extension services (last year) (%) | 91.9 | 41.4 | *** |
| Uses manure (%) | 68.3 | 44.6 | *** |
| *Panel B. Agricultural production* | | | |
| Maize yield (kg/ha) | 2352.3 | 1813.6 | *** |
| Legume yield (kg/ha) | 798.1 | 755.1 | |
| Value of all crops harvested ('000 MWK) | 213.2 | 124.2 | *** |
| Value of maize harvested ('000 MWK) | 98.2 | 78.3 | *** |
| Value of legumes and nuts harvested ('000 MWK) | 41.0 | 29.9 | *** |
| Net agricultural income ('000 MWK) | 172.8 | 103.2 | *** |
| Value of harvest sold ('000 MWK) | 52.4 | 22.8 | *** |
| Percent of harvest sold (%) | 23.2 | 18.0 | *** |
| Simpson production diversity index | 0.41 | 0.31 | *** |
| *Panel C. Household food consumption* | | | |
| Per capita annual food expenditure ('000 MWK) | 67.2 | 55.3 | *** |
| Value of purchased food ('000 MWK) | 96.8 | 83.6 | *** |
| Value of food from own production ('000 MWK) | 148.6 | 91.1 | *** |
| Simpson household dietary diversity index | 0.62 | 0.64 | |
| Simpson household dietary diversity index for purchased foods | 0.67 | 0.63 | *** |
| *Observations* | *397* | *538* | *935* |

Note: Columns 1 and 2 report means and column 3 reports statistical significance (stat. sign.) from mean comparison tests.

* $p<0.1$, ** $p<0.05$

*** $p<0.01$. MWK: Malawian Kwacha.


consumption from own production as well as from purchases (Table 2, Panel C). Household dietary diversity based only on purchased food is also higher among beneficiaries. Food group-level summaries reported in supplemental S2 Table also show inter-group differences with beneficiaries having higher per capita consumption expenditure on ASF such as eggs, milk, and dairy products but lower consumption of fruits.

Beyond establishing linkages between on-farm production diversity and household dietary diversity, understanding the causal pathways through which more diverse production can lead to better diets is key for policy. The first -and direct- channel is through higher consumption of own produced food, while the second -and indirect- channel is through higher agricultural income and food purchasing power [57]. The strength of the latter channel depends on the availability of nutritious foods locally and their affordability.

In our case, the extent to which pre-existing differences between treated and control households shown in Table 1 mediate the interaction between production and dietary diversity plays an important role in shaping the causal relationship. Intermediate regression results reported in supplemental S3 Table show the effect of program participation, instrumented by area

within 15 minutes of travel from the homestead, on the mediator (Simpson's production diversity index) to be 0.41 controlling for other variables (S3 Table, Column 1a). The estimate of the effect of the mediator on the outcome, controlling for treatment and other factors, is also significant in both cases where Simpson's household dietary diversity is measured using all foods consumed inside the household (S3 Table, Column 1b) and purchased foods (S3 Table, Column 1c).

Placebo test results from reduced form regressions implemented to assess the validity of our instrument are presented in S5 Table. As expected, the coefficient of the instrument is significant both in the production and dietary diversity models for the treated group (as well as the whole sample) but not for the control group. As our approach assumes that the treatment is endogenous with respect to the final outcome (dietary diversity) due to confounders that also jointly affect agricultural production, these test results provide indirect support for the validity of the exclusion restriction. That is, the effect of the program (that has not benefited the control group) on dietary diversity operates exclusively through the program's effect on production diversity.

Results from the IV mediation analysis are reported in Table 3. Program participation increases the Simpson's household dietary diversity by about 0.29 with the indirect -or mediated- effect estimated at 0.38 (Table 3, Column 1). This mediated effect accounts for 132% of the total effect of program participation on household dietary diversity, hence it is partly offset by the negative direct effect of program participation on dietary diversity. Albeit the fact that

**Table 3. Impact of program participation on production diversity and dietary diversity.**

| | (1) | (2) |
|---|---|---|
| | **Simpson's dietary diversity (all food)** | **Simpson's dietary diversity (purchased food)** |
| Total effect | 0.290*** | 0.192*** |
| | (0.077) | (0.074) |
| Direct effect | -0.094*** | -0.033 |
| | (0.033) | (0.031) |
| Mediated (or indirect) effect | 0.384*** | 0.225** |
| | (0.130) | (0.113) |
| Observations | 935 | 935 |
| Kleibergen-Paap F-statistic for the excluded instruments in first stage one (T on Z) | 31.17 | 31.17 |
| Kleibergen-Paap F-statistic for the excluded instruments in first stage two (M on Z\|T) | 20.13 | 20.13 |
| Mediation effect as a percentage of the total effect (%) | 132.3 | 117.1 |

Note: Results from the IV mediation analysis are reported. Dependent variables in columns 1 and 2 are Simpson's dietary diversity indices based on all food consumed and purchased food consumed by the household, respectively. T is indicator for program participation. M is Simpson's production diversity index. Excluded instrument (Z) is area of household plots within 15 minutes of travel. Control variables include household size, age and gender of the household head, average years of adult education, number of adults and children in the household, indices for dwelling condition and durable agricultural assets, temperature, slope, precipitation, and indicators for self-reported experience of droughts and crop diseases. Parameter estimates of exogenous controls not shown as they are partialled out using the Frisch-Waugh-Lovell theorem for ease of estimation. Eicker-Huber-White standard errors reported in parentheses.

*** $p<0.01$

** $p<0.05$, * $p<0.1$


the mediated effect is larger than the total effect may appear counterintuitive, it has been noted that a positive total effect stemming from a positive (larger) mediated effect partly offset by a negative direct effect would be perfectly conceivable [19]. In our setting, for example, regardless of the production channel the direct effect could be negative likely owing to a substitution mechanism at play, for which an increase in dietary diversity driven by program participation is partially reduced via a decrease in consumption of other food groups. For instance, substantial time and labor investments in the production of crops targeted by the program may results in lower investments in the production, and consumption, of other food sources.

On the other hand, beneficiaries seem to purchase more diverse foods from the market, relative to the control group, highlighting the role of the indirect -or income- effect of program participation on household diets (Table 3, Column 2). Both total and mediated effect sizes when diversity is measured based on purchased foods only account for approximately two-thirds of the increase in the Simpson's index based on all foods consumed inside the household (Table 3, Column 1). This result suggests that the market channel is more important than higher food own-consumed generated by greater production diversity among beneficiary households in line with previous findings [4]. Moreover, the direct effect turns smaller and not statistically significant when the Simpson's index of purchased food items is used as dependent variable, suggesting that the substitution effect is associated to a reduction in consumption of other food crops produced by the household. These findings persist when we measure production and dietary diversity using Shannon's diversity index as shown in supplemental S4 Table.

To further examine pathways from program participation to dietary diversity, we re-estimate Eqs 5 and 6 using two alternative mediators (M) based on program target crops -the value of maize harvest per hectare and the value of pulse harvest per hectare. The mediated -or indirect- effect of program participation on Simpson's dietary diversity index based on all food consumed through maize harvest is significantly higher than that through pulse harvest (Table 4, Columns 1 and 2). When the Simpson's dietary diversity index based on purchased food is used as dependent variable, the mediation effect through maize harvest is still statistically significant, unlike the same effect through pulse harvest (Table 4, Columns 3 and 4). Perhaps more importantly, the direct effect is positive and significant when pulse harvest value is used as mediator variable, suggesting other important channels affecting overall positive total effect that are not captured by the mediator, such as maize income. Overall, these findings suggest that the positive impacts of the program -both through enhanced market-related purchasing power as well as through production-led increases in own consumption–on dietary diversity are primarily generated by a propulsive effect of program participation on maize profitability.

S1 and S2 Figs compare distribution of estimated probabilities of treatment (propensity scores) used to estimate ATT and ATE based on IPWRA. The distributions are more comparable after matching as depicted by both box and kernel density plots. Matching reduced Rubin's B from 56 to 12 and Rubin's R from 1.8 to 1, both in the recommended range. ATT and ATE estimates are reported in S6 Table. Only ATT is marginally significant (at 10% level) when dietary diversity is measured based on all foods consumed inside the household (columns 1 and 2), while both ATT and ATE are significant when diversity is measured based only on purchased foods (columns 3 and 4), although ATE is only marginally significant (column 4).

Stronger effect on the diversity of purchased foods is consistent with results from the IV mediation analyses in Table 3 where approximately two-thirds of the increase in the Simpson's dietary diversity index –computed based on all foods consumed inside the household– was due to increase in the diversity of food purchased from the market. On the other hand, the magnitude of impact estimates based on IPWRA is smaller than that from IV mediation analysis. While both IV and risk adjustment (RA) approaches such as IPWRA are designed to

**Table 4. Impact of program participation on dietary diversity, as mediated by the value of maize and pulse harvests.**

| | (1) | (2) | (3) | (4) |
|---|---|---|---|---|
| | Simpson's dietary diversity (all food) | | Simpson's dietary diversity (purchased food) | |
| Total effect | 0.299*** | 0.290*** | 0.201*** | 0.192*** |
| | (0.080) | (0.077) | (0.076) | (0.074) |
| Direct effect | -0.036* | 0.091*** | 0.001 | 0.075*** |
| | (0.020) | (0.031) | (0.019) | (0.027) |
| Mediation (or indirect) effect | 0.335*** | 0.199* | 0.200** | 0.116 |
| | (0.121) | (0.103) | (0.100) | (0.072) |
| Observations | 935 | 935 | 935 | 935 |
| Kleibergen-Paap F-statistic for the excluded instruments in first stage one (T on Z) | 31.16 | 31.17 | 31.16 | 31.17 |
| Kleibergen-Paap F-statistic for the excluded instruments in first stage two (M on Z\|T) | 27.71 | 17.31 | 27.71 | 17.31 |
| Mediation effect as a percentage of the total effect (%) | 111.9 | 68.56 | 99.31 | 60.68 |

Note: Results from the IV mediation analysis are reported. Dependent variable in columns 1 and 2 are Simpson's dietary diversity index based on all food consumed. Dependent variable in column 3 and 4 is Simpson's dietary diversity index based purchased food consumed by the household. M in columns 1 and 3 is per capita value of maize harvest in thousands of Malawi Kwacha (MWK). M in columns 2 and 4 is per capita value pulse harvest in thousands of MWK. T is indicator for program participation. PD is Simpson's production diversity. Excluded instrument (Z) is area of household plots within 15 minutes of travel. Control variables include household size, age and gender of the household head, average years of adult education, number of adults and children in the household, indices for dwelling condition and durable agricultural assets, temperature, slope, precipitation, and indicators for self-reported experience of droughts and crop diseases. Parameter estimates of exogenous controls not shown as they are partialled out using the Frisch-Waugh-Lovell theorem for ease of estimation. Eicker-Huber-White standard errors reported in parentheses.

*** $p<0.01$

** $p<0.05$

* $p<0.1$.


mitigate confounding bias in non-experimental methods for impact estimation, the two approaches are not directly comparable and IV estimates cannot be truly interpreted as ATT or ATE [58]. When treatment effects are homogeneous across the target population or, if heterogeneous, are unrelated to treatment assignment, RA and IV estimates produce comparable results when corresponding identifying assumptions hold. On the other hand, when treatment effects are heterogeneous and potentially related to treatment assignment, RA and IV approaches may produce asymptotically different estimates as has previously been noted [58–60].

Our findings on the limited increase in own consumption of pulses appear to be in line with the literature. Indeed previous evidence from Malawi shows that pulse consumption is relatively inelastic to both income and pulse production [61,62]. Pulses are highly income inelastic among urban Malawian households, and even inferior goods among better-off households, showing an expenditure elasticity close to unity among rural households which may be related to the high value of international pulses trade [62]. Contrary to expectations, a study has found [62] a decline in per capita pulses consumption in rural Malawi despite the country's large-scale FISP, where legume seeds are either subsidized or granted for free. In the program under analysis, beneficiaries are exposed to pulses-based technologies through demonstration field days, and therefore they might not have gained enough insights on their nutritional benefits.

Boosting agricultural production diversity is often considered a promising approach to improve dietary diversity for poor and vulnerable farmers, either by increasing availability of more nutritious food for subsistence-oriented smallholders, by enhancing their purchasing

power, or both. Nonetheless, existing empirical literature on the linkages between production and dietary diversity is ambiguous. Earlier evidence points out that, while improvements in agricultural production diversity -and productivity- are necessary to enhance access to food and rural household income, they may not be sufficient to ensure dietary diversity, because agricultural innovations that increase the production of high-value and nutrient-dense crops could yield limited effects on their consumption due to the persistence of dietary habits and limited nutritional awareness. Several factors mediate the interaction between production diversity and dietary diversity, including market access, awareness about the nutritional content of targeted agricultural commodities, and intra-household decision making that a sound policy can effectively contribute to shape. Evidence is also mixed on the role of the direct and indirect impact pathways. For example, while the association between production and dietary diversity was found to be due to the direct pathway in a study from Ghana [12], the income pathway was found to be relevant in a cross-country study that includes Malawi [4].

Each of these factors require different policy course. Our findings of higher maize yield, higher value of maize and pulse harvest, higher net agricultural income, and higher crop sale (in levels and as a share of total harvest) point towards the importance of increasing the productivity and profitability to enhance market purchasing power. Efforts aiming at reducing barriers to better integration into output markets (e.g., limited information about prices and high transportation costs) could enhance participation in profitable markets and boost household dietary diversity. On the other hand, these market-oriented actions should be complemented with an active soil fertility monitoring, aimed at increasing nutritional content of crops that are otherwise sourced from the market with a consequent welfare-decreasing effect. Hence, the initial specialization strategy in the main staple crop should be accompanied by a diversification strategy to the extent that the new legume crops attain desirable physical properties to be able to substitute market-sources commodities.

During this process, attention should be given to intra-household decision making regarding the production, marketing, and consumption of different commodities. For higher production diversity to translate into more diverse diets, it is crucial that both men and women are aware of the nutritional values of different commodities. Specific to Malawi, for example, one study finds that in households with both adult men and women, informational campaigns including about nutrition that jointly target men and women have a stronger effect on household food security relative to campaigns that target only one gender [63]. Embedding adequate gender considerations will be especially important given the increasing role Malawian men are playing in decisions about food purchases as well as food preparation [64].

Our findings highlight once more that regardless of the strength of the linkages between production and dietary diversity, sequential actions in food policy would be necessary to address a host of complementary, and not contrasting, factors to maximize household-specific comparative advantages in crop cultivation, with the additional benefits of adoption of new crops and varieties that further increase the productivity and profitability of crops already grown by the household. The mediation role of pulses' adoption driven by the program under analysis is indeed a case in point for boosting crop diversification policy.

## Conclusion

We investigate the statistical associations between production diversity and dietary diversity in Malawi using cross-sectional household survey data collected as part of an environmentally sustainable agricultural intensification research program from program beneficiaries and a random sample of non-beneficiary, and pure control households. Program beneficiaries test various innovations and practices, including sole maize with different fertilizer application

rates, manure application, multiple legumes, maize–legume intercropping, and intercropping between two legumes.

Descriptive evidence shows that program beneficiaries were systematically different from control households along several dimensions considered. They were also able to attain higher value of agricultural production and net agricultural income as well as more diverse production, compared to the control group. Considering the systematic differences documented between treated and control groups, we employ an instrumental variables (IV) mediation analysis framework to estimate the impact of program participation on household dietary diversity. While traditional IV framework is used to identify unbiased impact estimates based on observational data, it does not allow us to unpack causal impact when both treatment -program participation in our case- and an intermediate outcome -on-farm production diversity-, jointly cause a secondary outcome -household dietary diversity.

Results point to a positive and significant impact of the program on dietary diversity, which is mainly driven by the increase in production diversity of beneficiary households. In sum, participation led to a more diverse agricultural production, and, through this mediating channel, to an increase in dietary diversity. However, this increase does not seem to be primarily related to cultivation of pulses, the main crop group targeted by the program, which are rich in crucial micronutrients and contain better and higher quality protein than other grains. Rather, positive effects are mostly associated to maize production, a secondary program targeted crop. Enhanced maize production boosts both own-consumption and, perhaps more importantly, agricultural income, allowing households to purchase and consume more diverse food items.

We document a weak association between production and own consumption of pulses, underscoring the importance of complementing production-oriented programs with demand-side interventions to promote nutritional awareness. The positive and significant correlations between production diversity and diversity of purchased food highlights the importance of access to food markets for increasing and reinforcing nutritional gains associated to enhanced on-farm production diversity. While our study provides useful insights on the linkages between agriculture and nutrition, it does not address potential intrahousehold reallocation and inequalities in food consumption. Also, diets may shift over time, especially due to nutritional education, albeit only in the medium to long-term. Also, the effects we document may be limited to the short-run due to limited time elapsed between beginning of the program and data collection -just one completed cropping season-.

Lastly, some caveats are necessary regarding the interpretation of our findings. While the novel empirical approach adopted allows us to overcome econometric challenges related to selection bias, endogeneity, simultaneity, and the cross-sectional nature of the data, the results provided here should nonetheless be interpreted with caution: the significant associations and correlations found are suggestive of causal relationships, but causal claims will have to be more thoroughly supported by new empirical evidence when follow-up data finally become available. Therefore, a key area for future research is a longitudinal analysis of longer-term effects of the program to examine whether and to what extent gains in knowledge, adoption of improved innovations, and environmental services can bring about longer-term effects on production and consumption patterns of beneficiary households.

## Supporting information

**S1 Fig. Box plots of propensity scores before and after matching.**
(TIF)

**S2 Fig. Kernel density plots of propensity score before and after matching.**
(TIF)

**S1 Table. Value of agricultural production by food group.**
(XLSX)

**S2 Table. Values of consumption expenditure by food group.**
(XLSX)

**S3 Table. Intermediate (first stage) results from IV mediation analysis.**
(XLSX)

**S4 Table. Impact on production diversity and dietary diversity (by diversity index).**
(XLSX)

**S5 Table. Reduced form regressions for production and dietary diversity.**
(XLSX)

**S6 Table. ATT and ATE effects on household dietary diversity (IPWRA).**
(XLSX)

## Acknowledgments

We are grateful to Viviana Celli for her valuable comments on earlier versions of the manuscript and Malawi Africa RISING project partners, especially Dr. Regis Chikowo, for their support during data collection. We thank Arkadeep Bandyopadhyay and Cleo Roberts for their contribution during data processing.

## Author Contributions

**Conceptualization:** Carlo Azzarri, Beliyou Haile.

**Data curation:** Beliyou Haile.

**Formal analysis:** Beliyou Haile, Marco Letta.

**Funding acquisition:** Carlo Azzarri.

**Methodology:** Carlo Azzarri, Beliyou Haile, Marco Letta.

**Project administration:** Carlo Azzarri.

**Writing – original draft:** Beliyou Haile, Marco Letta.

**Writing – review & editing:** Carlo Azzarri, Beliyou Haile, Marco Letta.

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
