## [Decision Letter · Decision Letter 0]

26 Nov 2021

PONE-D-21-30466Plant different, eat different? Insights from participatory agricultural researchPLOS ONE

Dear Dr. Haile,

Thank you for submitting your manuscript to PLOS ONE. After careful consideration, we feel that it has merit but does not fully meet PLOS ONE’s publication criteria as it currently stands. Therefore, we invite you to submit a revised version of the manuscript that addresses the points raised during the review process. The topic is very interesting and timely especially now that the 2021 Nobel prize for economics has been awarded to scientists and their research focusing on selection biases. The reviewers raise some questions on the methodologies that need some further clarification on your part. 

We look forward to receiving your revised manuscript.

Kind regards,

Gideon Kruseman, Ph.D.

Academic Editor

PLOS ONE

Journal Requirements:

3. Please update your submission to use the PLOS LaTeX template. The template and more information on our requirements for LaTeX submissions can be found at http://journals.plos.org/plosone/s/latex

4. We note that you have referenced (ie. Bewick et al. [5]) which has currently not yet been accepted for publication. Please remove this from your References and amend this to state in the body of your manuscript: (ie “Bewick et al. [Unpublished]”) as detailed online in our guide for authors

Reviewers' comments:

Reviewer's Responses to Questions

**Comments to the Author**

1. Is the manuscript technically sound, and do the data support the conclusions?

Reviewer #1: Yes

Reviewer #2: Yes

Reviewer #3: Partly

2. Has the statistical analysis been performed appropriately and rigorously? 

Reviewer #1: Yes

Reviewer #2: I Don't Know

Reviewer #3: Yes

3. Have the authors made all data underlying the findings in their manuscript fully available?

Reviewer #1: No

Reviewer #2: Yes

Reviewer #3: Yes

4. Is the manuscript presented in an intelligible fashion and written in standard English?

Reviewer #1: Yes

Reviewer #2: Yes

Reviewer #3: Yes

5. Review Comments to the Author

Reviewer #1: Reviewer’s Comments

The manuscript, is an interesting piece of research. This paper analyses the causal effects of on-farm production diversity on household dietary diversity in Malawi based using microdata collected as part of a participatory agricultural intensification program. This is a relevant topic in the body of literature particularly due to the fact that nutrition plays such a significant role human capital development and consequent productivity. Poor nutrition especially for children has long-lasting consequences to the society. After reviewing this paper, I have the following observations concerning the suitability of the article for publishing in the PLOS ONE journal.

In what follows I detail my suggestions for further improvement of the paper.

Abstract

Needs minor improvements.

1. The word commodities in the sentence “Among other innovations, the program aims to promote the production of pulses, commodities that are rich in micronutrients, have better-quality protein, and have nitrogen fixation benefits compared to other grains” can be removed with no alteration in meaning.

2. In the sentence “We find a significant positive treatment effect on dietary diversity, led by an increase in production diversity.” The treatment should be stated directly instead of letting the reader figure it out.

Introduction

3. “An important lesson from these studies is that while a diversification strategy has the potential to improve diets and nutrition, the strength of the association is quantitatively controversial.” In this statement authors have not stated what these quantitative controversies and makes the consequent paragraph on filling in this gap invalid. Authors should clear state what these controversies are and how their study solves those controversies if not justify why and how their study cannot be counted as another adding more controversy.

4. The statement “while 19% and 40% of the country’s population and children below the age of five years are estimated to be undernourished and stunted, respectively” is not clear whether the figures refer respectively to country population and children or undernourished and students.

5. The sentence “Analysis of the linkage between production diversity and dietary diversity in a crosssectional setting is challenging due to a host of confounding factors …….”

Should be moved from its current paragraph and be used as an opening sentence in the next paragraph

6. The finding that “While production diversification can lead to more diverse diets, increasing production of more nutritious crops such as pulses may not necessarily translate into their greater own consumption, due to the persistence of dietary habits or other location specific factors” should be more elaborately explained; why then should increase in income change dietary diversity despite “the persistence of dietary habits or other location specific factors”

Materials and methods

7. Authors need to explain why they have excluded group 2 in the analysis. Obviously if groups 1 and 2 were targeted and only those in group one engaged in the project then you have self-selection. But then the random (as stated) targeting with non-compliance only on the treatment side could provide the Intention To Treat (ITT) which is of interest. In case of non-compliance on both the treatment and the control side the random (as stated) targeting could be used as a clean instrument for participation in the program.

8. Given the nature of the instrument variable (Z) that is used then it makes sense if group 2 is included in the analysis. Because if only groups 1 (program participant) and group 3 (control) are used then there is no way the instrument will be explaining participation (by design) since the control group did not get exposure to the program.

9. Second paragraph of page 6, on-farm production should be removed “Diversity in on-farm production and household diets is measured based on the Simpson’s diversity”

10. There are different measures of production and dietary diversity, authors should argue why they chose to use the Simpson’s index over others.

11. The use of values (instead of quantities) to construct the Simpson’s Index is prone to being affected by food items with high values; authors should also explain why we should not be worried about this. In addition they should also explain how these values were obtained (market price? How about those which were not purchased/sold?)

Results and Discussion

12. A number of variables presented in Table 1 could have been affected by the program despite the claim by authors that these variables are unlikely to be affected by the program. For example wealth (measured by assets), dwelling quality, all agricultural inputs and practices variables. Since the program targeted agricultural production, obviously agricultural practices will be affected, incomes and wealth accumulation.

13. Page 14, monitory � monetary

14. That the claim is the effect ion dietary diversity is mainly through the indirect- channel is through higher agricultural income and food purchasing power. It would be crucial to show empirically whether the program had an effect on household income.

Reviewer #2: The paper touches a timely and relevant topic. Although the connection of production and consumption diversity has attracted some attention in the rural development literature, findings on the relationship are mixed. The paper therefore provides a valuable contribution to the existing body of literature. Despite its thematic appeal and well-presented structure - in view of this review - the manuscript should undergo some major revisions to be considered for publication. Main concerns include the claim of causality when interpreting regression results throughout the paper, the suitability of the selected instrument in the regression framework, the lack of background information on the Africa RISING program, as well as the absence of a comprehensive review of the literature on the linkages of production and consumption diversity.

Please find more detailed comments below. Major comments are listed first, minor comments are listed further below. (Line numbering of the manuscript would have been appreciated).

Major comments:

1. Instrumental variable is not convincing. Authors use area of agricultural land within 15 minutes of homestead to instrument household participation in the program. Authors do not argue convincingly why they expect agricultural land within 15 minutes of the homestead to only influence production diversity through program participation. Cropping patterns may differ with increasing distance from a plot to the homestead, e.g., households may cultivate more diverse crops on their plots closer to the homestead - irrespective of program participation. The authors do not present data on production diversity in close to homestead plots for beneficiaries and control households. Authors also do not present information on how the program aimed at increasing production diversity among its beneficiary households. More background information would be needed to support the argumentation for the instrument of choice. In its current narrative, the selection of the IV is not convincing (see also further comments on the IV selection below).

2. Claims of causality. The authors employ IV mediation analysis, which is an appealing tool, especially in the context of analysis of complex interactions - here participation in a program, agricultural production diversity, and household dietary diversity. Through its mediation framework, the tool allows the analysis of different pathways from the treatment to the outcome. Authors also provide different mediators and outcomes to complement their analysis. However, in several parts of the paper, authors claim causality. In context of the complex relationship and the previous comment on IV selection, authors should consider to be more cautious in interpretation of results (e.g., by speaking of associations/correlations instead of causation).

3. Framing of the story. The authors frame the story around PP – PD – DD (program participation- production diversity-dietary diversity). Yet, authors results (and interpretation thereof) indicate that positive associations between PP and DD are not necessarily associated with PD per se, but with increases in maize output and income, which in turn is used to increase HH DD. Authors claim in some parts of the manuscript that PD leads to higher DD, which should be rephrased to avoid misinterpretation.

4. Lack of background information on Africa RISING. Although the program is introduced briefly in the paper, the authors may want to provide more information on Africa RISING. In its current form, reader does not learn what the program does in terms of promotion of legume and/or maize production, when interventions started, from when the data presented stems, how beneficiaries were selected (targeted?) into the program, intervention logic, and modes of implementation. The presentation of program objectives, mode of beneficiary selection, location, timing, and concrete interventions are essential to contextualize the findings of regression results.

5. Lack of comprehensive review of literature examining linkages between production and consumption diversity. Although authors briefly refer to existing literature on the topic (e.g., last sentence of last paragraph of introduction p.3 and last sentence of following paragraph, p.3), the manuscript does not provide a comprehensive review of their findings. Authors should include such a review – also highlighting that findings are inconclusive and inconsistent, also pointing to the fact that the relationship (production diversity and consumption diversity) is likely to be highly complex and context dependent.

Minor comments (in order of appearance in the manuscript- line numbering would have been appreciated):

6. P.4 “We tackled these challenges by using household survey microdata […]”. Authors correctly point to the challenges associated with the examination of linkages between production and consumption diversity in the previous sentence. The cited sentence claims to present how these problems are tackled, yet it only presents the data source.

7. P.5 “In this study we use recently developed mediation analysis […]”. Authors may want to include a reference in relation to the method, as well as references in which the method has been applied in similar contexts

8. P.5 “The underlying mechanism, however, appears to be through higher purchase of more diverse food […]” and “[…] these effects are primarily due to an increase in the production of a secondary target crop, maize, and not to higher pulse production.” Considering this finding, authors may want to rephrase statements in the manuscript which may be misinterpreted by readers in the sense that diversification of production leads to diversification in production. E.g., abstract: “These findings highlight that, while diversifying production can lead to more diverse diets […]”.

9. P.5 Before Materials and Methods section, authors may consider providing background information on Africa RISING (also see comment #4).

10. P.6 “This study analyzes data […]”. Please explain why group 2 households were not considered in the analysis.

11. P.6 “Agricultural production data refer to […]”. Please briefly discuss advantages and disadvantage of 7-day recall approach. Please also reflect on how the harvest cycle/period of cultivated crops coincides with the concrete dates of the recall period. That is can households consume own produce in the 7-day period prior to interview.

12. P.7 Identification. Please define T, M and Y as used in the manuscript, when mentioned for the first time.

13. P.8 “[…] validity, exogeneity, and strength [of the IV]”, replace by relevance, exclusion restriction, and independence assumption?

14. P.9 “We posit the partial identifying assumption to hold in our setting, […]”. This assumption merits further discussion. e.g., what variables do authors expect to jointly affect T and PD (to confound DD through PD)? Why do authors expect other confounders of DD (not through PD) to be of less importance?

15. P.9-10 “[…] we estimate Equations 5 and 6 using 2SLS […]”. Considering the complex relationship between PD and DD, and the IV at hand, estimation of the causal chain seems quite ambitious. Authors may consider referring to associations instead of causal chains.

16. P.10 “Our choice of an exogenous instrument for program participation is guided by program design, […]”. As noted in an earlier comment, the manuscript does not contain much information on program design. Please include, also to support your argument for the IV.

17. P.10 “Specifically, we use total land area operated by the household […]”. It would be interesting to see whether there are differences in cropping patterns on these plots between participants and non-participants.

18. P.10 “[…] land area within 15 minutes travel would not affect DD unless through program participation.” This is a strong statement and goes along with major comment #1. There may be channels other than program participation through which land area within 15 minutes of homestead may affect DD. E.g., if households tend to cultivate certain crops more often on plots closer to their home, and if these crops tend to be marketed/ consumed more often, DD could also be affected by (pre-existing) cropping patterns on these plots, irrespective of program participation.

19. Page 11 The reference to Fig. 1 appears twice.

20. P.12 “Since survey data were collected […]. Please indicate when data was collected and when program activities started.

21. P. 13 Table 1. Number of observations (n=935) does not match with the total number of observations presented in the data section (N=1149). Please explain.

22. P. 14 Table 2. Table 2 shows that beneficiaries tend to have higher maize yields, higher value of maize harvested, and higher net agricultural income with relative importance of income from maize >> income from legumes. How does maize production fit into the assessed relationship between production diversity and dietary diversity? From the manuscript, it seems like the program mainly tries to introduce additional legumes into household production. What is the role of maize production?

23. P.15 “As noted in previous studies, […]”. The authors may want to contextualize their findings regarding the sources included in the cited literature (e.g., sources 1-7). E.g., literature analyzing linkages between on -farm production diversity and household consumption diversity has produced mixed findings: Ecker (2018) finds that production diversity is associated with HH consumption diversity mainly through direct pathways (HH consuming own produce), while Sibhatu et al. (2015) find that increase in HH consumption diversity is associated with increased HH income (and not by consumption of own produce).

24. P.16 “This result suggests that the market channel is more important […]”. Please also put into context of literature on PD and DD.

25. P.22 “Results point to an overall positive impact of the program on dietary diversity […]”. In the context of results presented earlier (direct effect being outsized by indirect effect), this formulation may be misleading. Consider rephrasing.

26. P.22 “Enhanced maize production boosts both own-consumption and, perhaps more importantly, agricultural income, allowing households to purchase and consume more diverse food items.” Would this finding not imply that PD does not play a large role in DD. After all, Maize is the most grown cash crop in Malawi, and as such not associated with PD (or is it?). How would T (program participation)->M (production diversity)->Y (dietary diversification) hold in this case? What does the program promote in terms of maize production? Does PD even matter?

27. Supplementary Material. Results from models S1(1b) and Table 3 (1) are not the same, which they should (same T, M, and Y). Please explain, also why number of observations is not the same for both models (S1 n=931, table 1 n=935).

Reviewer #3: Overall:

The study design (cross-sectional survey) does not seem to adequately measure program participation, and it seems unlikely that there would be significant impacts on production after just 1 year of the study, so the overall results are not very compelling or convincing. The authors need to provide more information about the intervention and study design to be able to justify their conclusions. Further, there are several recent studies in Malawi that are not included in the paper and the authors reference to this as an ‘agricultural intensification’ study seems at odds with the diversification emphasis. Finally, gender dynamics in agriculture need to be adequately addressed in the paper.

Introduction:

- The authors conclude from their brief review of the linkages between diversified farming systems and nutritional outcomes is that ‘An important lesson from these studies is that while a diversification strategy has the potential to improve diets and nutrition, the strength of the association is quantitatively controversial.’ [italics added]. My understanding of the literature is somewhat different from this conclusion. Recent reviews of a considerable literature on this subject have all shown a positive, significant association between production diversity and dietary diversity for farming households, including 2 reviews included in the literature review (Jones 2017; Sibhatu et al. 2015). The question is perhaps to what extent other factors matter more or mediate that relationship, and several studies have pointed to the significance of factors such as gender relations in mediating that relationship, including studies done in Malawi (e.g. Bezner Kerr et al. 2019).

- The authors refer to their intervention as ‘agricultural intensification’ but from the description the intervention appears to be crop diversification. Crop/farm diversification is often contrasted to intensification in the literature, because intensification usually entails monocropping and increased input use rather than diversification. Diversification, in contrast, or diversified farming systems, is often linked to an agroecological approach and has been a subject of policy debate for farming and food systems because of many environmental services as well as food security and nutritional outcomes. This broader debate within which diversification of food systems sits might be briefly noted in the literature review. Some possible references:

o Bezner Kerr, R. et al. 2021. Can agroecology improve food security and nutrition? A review. Global Food Security 29. https://doi.org/10.1016/j.gfs.2021.100540

o Rasmussen, L.V., et al. 2018. Social-ecological outcomes of agricultural intensification. Nat. Sustain. 1, 275–282. https://doi.org/10.1038/s41893-018-0070-8

- The authors could integrate more recent studies that examine these relationships in Malawi or similar contexts that were not included in this review:

Madsen, S. et al. 2021. Explaining the impact of agroecology on farm-level transitions to food security in Malawi. Food Security 13: 933–954.https://doi.org/10.1007/s12571-021-01165-9

Kansanga, M.M. et al. 2021. Agroecology and household production diversity and dietary diversity: Evidence from a five-year agroecological intervention in rural Malawi. Social Science and Medicine 288, 113550. https://doi.org/10.1016/J.SOCSCIMED.2020.113550

Santoso, M.V. et al. 2021. A nutrition-sensitive agroecology intervention in rural Tanzania increases children’s dietary diversity and household food security but does not change child anthropometry: results from a cluster-randomized trial. Journal of Nutrition.

https://doi.org/10.1093/jn/nxab052

Bezner Kerr, R. et al. 2019. Participatory agroecological research on climate change adaptation improves smallholder farmer household food security and dietary diversity in Malawi. Agriculture, Ecosystems and Environment 279: 109-121. https://doi.org/10.1016/j.agee.2019.04.004

Snapp, S. S.,&Fisher, M. (2015). "filling themaize basket" supports crop diversity and quality of household diet in Malawi. Food Security, 7(1), 83–96.

Methods:

There is very limited information about the intervention. Who was involved, and what support was provided to the household? How was it participatory?

How was participation measured in the study – simply that they were enrolled? In which case how was lack of participation taken into account? Other mediation studies have much more detailed information about the intervention in order to determine the overall relationships, eg. Cetrone et al. 2021. Food security mediates the decrease in women's depressive symptoms in a participatory nutrition-sensitive agroecology intervention in rural Tanzania. Public Health Nutr. 24(14):4682-4692. doi:10.1017/S1368980021001014. Epub 2021 Mar 12. PMID: 33706829.

Results

Given that this study took place only 1 growing season after the intervention began, it seems somewhat premature to anticipate impacts on dietary diversity from production diversity – it may take 2 or more growing seasons for farmers to realize some of the longer term impacts from diversified production systems, including production as well as improved soil fertility, reduced pests and diseases etc. In addition they may not be able to produce significant enough yield of the new crop in the first season to generate enough for sale, seed and consumption and so may prioritize one of those options. These aspects need to be discussed in relation to your results.

The authors state that ‘This result suggests that the market channel is more important than the increase in food own-consumed generated by enhanced production diversity..’. This is a cross-sectional study, so the survey results do not demonstrate that there has been ‘enhanced production diversity’ only that participants grow more food, they might have been doing so prior to the intervention.

The authors also seem to somewhat misrepresent the findings of a review on agricultural diversification and dietary diversity. They indicate that “on-farm diversity has a small and nonlinear association with dietary diversity” and cite Jones (2017). The nonlinear association is true, pointing to several different pathways discussed in the review by Jones (2017), but note that the author found that: “agricultural biodiversity has a small but consistent association with more diverse household- and individual-level diets, although the magnitude of this association varies with the extent of existing diversification of farms. Greater on-farm crop species richness is also associated with small, positive increments in young child linear stature.”

Discussion

The authors seem to mention gender issues as an afterthought at the end of the section, and only in relation to involving both men and women. Previous research in Malawi and in other places has demonstrated that unequal gender dynamics within households can also influence dietary outcomes, such that increased production diversity may not translate into consumption of the food products (see Ruel and Alderman 2013 for example). Addressing gender dynamics as part of a nutrition-sensitive intervention has also been shown to increase the likelihood of improved dietary diversity (e.g. Bezner Kerr et al. 2019; Santoso et al. 2021). this aspect of gender inequality is not discussed in the paper and does not seem to have been taken into account adequately in the study design or analytical methods.

Ruel MT, Alderman H. Nutrition-sensitive interventions and

programmes: How can they help to accelerate progress in improving

maternal and child nutrition? Lancet 2013;382(9891):536–51.

The authors state that “Our results point towards the importance of increasing the productivity and profitability ‒ including of Malawi’s main staple crop maize ‒ to enhance market purchasing power.” Please explain how your results point to the importance of increasing maize production – given the dominance of maize in production and consumption, it is not clear how your results point to such a finding.

They also state that “Efforts aiming at reducing barriers to better market integration (e.g., limited information about prices and high transportation costs) could enhance participation in profitable markets and boost household dietary diversity.” The results presented do not appear to say anything about market integration, so it is not clear how these conclusions are drawn. Could not better market integration increase consumption of unhealthy imported purchased foods, as observed in many other parts of the Global South?

The authors go on to state: “On the other hand, these market-oriented actions should be complemented with an active soil fertility monitoring, aimed at increasing nutritional content of crops that are otherwise sourced from the market with a consequent welfare-decreasing effect. Hence, the initial specialization strategy in the main staple crop should be accompanied by a diversification strategy to the extent that the new legume crops attain desirable physical properties to be able to substitute market-sources

commodities.”

The results presented do not appear to provide any information about soil fertility or nutritional content of crops, so again it is not clear how such conclusions are drawn.

It is not clear how an intensification strategy would be accompanied by a diversification strategy since these often work at cross purposes, as a number of studies of the FISP program demonstrates:

Chibwana, C., Fisher, M., Shively, G., 2012. Cropland Allocation Effects of Agricultural Input Subsidies in Malawi. World Dev. 40, 124–133. https://doi.org/10.1016/j.worlddev.2011.04.022

The authors note that this is surprising, but based on my own knowledge and understanding of FISP from long-term research in Malawi, while legumes are officially a part of the program, they constitute a much smaller proportion of the total amount of seeds distributed, and the emphasis is on maize production.

6. PLOS authors have the option to publish the peer review history of their article (what does this mean?). If published, this will include your full peer review and any attached files.

Reviewer #1: No

Reviewer #2: No

Reviewer #3: No

---

## [Author Response · Author response to Decision Letter 0]

13 Jan 2022

Please see the attached response to the three reviewers.

---

## [Decision Letter · Decision Letter 1]

9 Feb 2022

PONE-D-21-30466R1Plant different, eat different? Insights from participatory agricultural researchPLOS ONE

Dear Dr. Haile,

Thank you for submitting your manuscript to PLOS ONE. After careful consideration, we feel that it has merit but does not fully meet PLOS ONE’s publication criteria as it currently stands. Therefore, we invite you to submit a revised version of the manuscript that addresses the points raised during the review process.

Reviewer 2 still has a number of questions related to your use of instrumental variables. 

We look forward to receiving your revised manuscript.

Kind regards,

Gideon Kruseman, Ph.D.

Academic Editor

PLOS ONE

Reviewers' comments:

Reviewer's Responses to Questions

**Comments to the Author**

1. If the authors have adequately addressed your comments raised in a previous round of review and you feel that this manuscript is now acceptable for publication, you may indicate that here to bypass the “Comments to the Author” section, enter your conflict of interest statement in the “Confidential to Editor” section, and submit your "Accept" recommendation.

Reviewer #1: All comments have been addressed

Reviewer #2: (No Response)

Reviewer #3: All comments have been addressed

2. Is the manuscript technically sound, and do the data support the conclusions?

Reviewer #1: Yes

Reviewer #2: Yes

Reviewer #3: Yes

3. Has the statistical analysis been performed appropriately and rigorously? 

Reviewer #1: Yes

Reviewer #2: Yes

Reviewer #3: Yes

4. Have the authors made all data underlying the findings in their manuscript fully available?

Reviewer #1: Yes

Reviewer #2: Yes

Reviewer #3: Yes

5. Is the manuscript presented in an intelligible fashion and written in standard English?

Reviewer #1: Yes

Reviewer #2: Yes

Reviewer #3: Yes

6. Review Comments to the Author

Reviewer #1: The authors have addressed all the comments well and provided their responses clearly.

I congratulate them for that.

Reviewer #2: Thanks to the authors for their detailed responses to the comments. Most of the remarks made during the first review have been fully addressed. In particular, the manuscript now contains a section on program details, authors have altered wordings in relation to claims of causal relation and have linked their findings on the relation of production and dietary diversity to relevant literature. The manuscript continues to be well-structured, well-written and comprehensive in terms of data presentation and analysis.

Despite the reviews undertaken, the central comment during the first review – the choice of the instrumental variable - has only been addressed partially. Please find the main arguments against the IV in its current way of presentation below. Being fully aware of the complexity of discussions about IV selection, some suggestions to support the current IV are provided for the authors consideration.

1. Choice of IV.

It is acknowledged that authors have put great effort into the selection of an appropriate instrument for their analysis. The new manuscript contains additional tests (placebo test) in the supplementary material, and a comprehensive discussion of its results and implications for the analysis in the main body of the manuscript. Authors have also revised the Identification chapter and provided a theoretical background on IV choice in IV mediation analysis, based on which authors have motivated their IV choice. The arguments presented by the authors can be followed, and the instrument is certainly a suitable predictor for program participation (T) – as it was a precondition to be selected as beneficiary.

Yet the presented discussion and results are not (yet) conclusive regarding the overall suitability of the IV. This concerns mainly the exclusion restriction. Authors use land within 15 minutes of homestead to instrument program participation. In the context of smallholder agriculture, land holdings and locations of land holdings may be correlated with several (unobserved) factors that may simultaneously also influence crop and livestock production patterns, management practices, yield levels, output marketing, all of which ultimately may influence HH consumption decisions. In its current form, the manuscript does not show convincingly that treatment and control households do not differ systematically with respect to some unobserved features that influence both size of land holdings within 15 minutes of the homestead (the precondition to receive treatment) and the outcome (dietary diversity). This is not trivial, as descriptive comparison of treatment and control groups presented in tables 1 and 2, shows significant differences across several variables, e.g., treatment households tend to be better educated, less female led, have larger total land holdings, are less likely to be poor, and own more livestock. While it can be argued that certain differences are associated with the treatment itself, especially Simpson’s production diversity index and marketed value of legumes, other differences such as households’ land holdings, location of landholdings, and poverty incidence (based on both durable non-agricultural assets, and on dwelling condition) are likely independent of the treatment and therefore pre-existing. Thus, it cannot be ruled out that households with relatively more land holdings within 15 minutes of the homestead (essentially the treatment households) systematically share unobserved characteristics that affect their wealth status, consumption expenditures and dietary diversity. In this case, estimates of treatment effect on DD would be biased, most likely upward biased.

Authors have responded, with inter alia, a placebo test. As stated by the authors, this test provides indicative evidence that the size of land holdings within 15 minutes of the homestead seems to affect production diversity only through program participation (treatment) as the reduced form model yields an insignificant coefficient for the regression of PD on landholdings within 15 minutes of the homestead among control households. Thus, land holdings as such are not likely to directly affect production diversity of a given household. Yet, the results do not show whether there are any links between landholdings within 15 minutes of the homestead and dietary diversity other than through production diversity. In other words, while the test provides strong evidence for the instrument’s relevance, it does not provide support for the exclusion restriction to hold (the main critique of this comment).

To provide additional support for their choice of IV, authors may consider running the placebo test directly on the outcome (not on the mediator) to test whether the exclusion restriction holds (IV only affects DD through PD). For the treatment group the coefficient of such a test would indicate the “[..] effect of T on Y that operates exclusively through its effect on M” (citation from manuscript p.14). For the control group, the coefficient should not be statistically different from zero, as the only way through which landholdings close to the homestead would affect DD should be through PD (T). Results could support the statement made on p. 16 “we can safely assume that T is endogenous with respect to DD

due to confounders that affect DD primarily through PD”.

In addition, authors may consider running a (supplementary) propensity score matching, using available control variables, especially land holdings, wealth status, as matching parameters to estimate the ATE and ATT of the treatment on dietary diversity. Although bias due to unobserved pre-treatment differences may still not be ruled out, PSM would ensure that treatment effects are assessed based on similar HH in control and treatment groups, and magnitude of results could be compared with those of IV mediation analysis. In this context, authors mays consider using sensivity analysis to test for unobserved confounders.

Additional minor comments

2. Introduction and study setting.

Authors may consider sharpening the introduction and study setting chapters, by focusing on the relationship of diversification of agricultural production and household nutrition. Especially the opening sentences of the study setting chapter feel out of context (although content is certainly relevant for the study). Authors may consider opening the study setting chapter with an overview of the status of farms’ production diversity (or lack thereof) and continue with a description of how the objectives of the program under study aim at enhancing production diversity and what are expected outcomes (e.g., diversifying maize production).

3. Page 6 first paragraph.

The preceding paragraph states that findings are mixed across and within studies. It is not evident how the cited examples for South America support this contradiction of findings (tuber are promoted and consumed more often; livestock ownership leads to greater animal sourced food consumption).

Reviewer #3: The authors have addressed my concerns and comments. The paper provides a useful and relevant paper in this topical area.

7. PLOS authors have the option to publish the peer review history of their article (what does this mean?). If published, this will include your full peer review and any attached files.

Reviewer #1: No

Reviewer #2: No

Reviewer #3: No

---

## [Author Response · Author response to Decision Letter 1]

22 Feb 2022

Please see the attached response.

---

## [Decision Letter · Decision Letter 2]

11 Mar 2022

Plant different, eat different? Insights from participatory agricultural research

PONE-D-21-30466R2

Dear Dr. Haile,

We’re pleased to inform you that your manuscript has been judged scientifically suitable for publication and will be formally accepted for publication once it meets all outstanding technical requirements.

Kind regards,

Gideon Kruseman, Ph.D.

Academic Editor

PLOS ONE

Additional Editor Comments (optional):

Reviewers' comments:

Reviewer's Responses to Questions

**Comments to the Author**

1. If the authors have adequately addressed your comments raised in a previous round of review and you feel that this manuscript is now acceptable for publication, you may indicate that here to bypass the “Comments to the Author” section, enter your conflict of interest statement in the “Confidential to Editor” section, and submit your "Accept" recommendation.

Reviewer #2: All comments have been addressed

2. Is the manuscript technically sound, and do the data support the conclusions?

Reviewer #2: Yes

3. Has the statistical analysis been performed appropriately and rigorously? 

Reviewer #2: Yes

4. Have the authors made all data underlying the findings in their manuscript fully available?

Reviewer #2: Yes

5. Is the manuscript presented in an intelligible fashion and written in standard English?

Reviewer #2: Yes

6. Review Comments to the Author

Reviewer #2: I thank the authors for their additional work undertaken in relation to the choice of instrumental variable. The discussion and supplementary analysis now more comprehensively support the IV used in the analysis. There are no further comments.

7. PLOS authors have the option to publish the peer review history of their article (what does this mean?). If published, this will include your full peer review and any attached files.

Reviewer #2: No

---

## [Editor Report · Acceptance letter]

16 Mar 2022

PONE-D-21-30466R2 

Plant different, eat different?  Insights from participatory agricultural research 

Dear Dr. Haile:

I'm pleased to inform you that your manuscript has been deemed suitable for publication in PLOS ONE. Congratulations! Your manuscript is now with our production department. 

Kind regards, 

on behalf of

Dr. Gideon Kruseman 

Academic Editor

PLOS ONE